# Sequences of Alterations in Inflammation and Autophagy Processes in Rd1 Mice

**DOI:** 10.3390/biom13091277

**Published:** 2023-08-22

**Authors:** Javier Martínez-González, Ángel Fernández-Carbonell, Antolin Cantó, Roberto Gimeno-Hernández, Inmaculada Almansa, Francisco Bosch-Morell, María Miranda, Teresa Olivar

**Affiliations:** Department of Biomedical Sciences, Faculty of Health Sciences, Institute of Biomedical Sciences, Cardenal Herrera-CEU University, CEU Universities, 46115 Valencia, Spain; javier.martinezgonzalez@uchceu.es (J.M.-G.); angel.fernandez.carbonell@gmail.com (Á.F.-C.); antolin-cantocatala@uchceu.es (A.C.); rgimenohdez@gmail.com (R.G.-H.); ialmansa@uchceu.es (I.A.); fbosch@uchceu.es (F.B.-M.); mmiranda@uchceu.es (M.M.)

**Keywords:** retinitis pigmentosa, microglia, macroautophagy, chaperone-mediated autophagy

## Abstract

**Simple Summary:**

In this work, we have demonstrated alterations in microglia and macroautophagy in rd1 mice (one retinitis pigmentosa (RP) model) at the first stages of the disease (when the rods are dying). When there are almost no rods, and the cones are dying, chaperone-mediated autophagy alterations (CMA) are found in RP retinas. Based on our results, it would be reasonable to conclude that inflammation and macroautophagy processes could be possible alternatives in the treatment of RP, in the initial stages. In this phase, cones, which are mainly responsible for human vision, have not yet degenerated, thus allowing a very high quality of life for patients if retinal degeneration can be stopped or slowed down in this phase. On the other hand, CMA would constitute a possible therapeutic target later, when cones are degenerating.

**Abstract:**

(1) Background: the aim of this work was to study microglia and autophagy alterations in a one retinitis pigmentosa (RP) model at different stages of the disease (when rods are dying and later, when there are almost no rods, and cones are the cells that die. (2) Methods: rd1 mice were used and retinas obtained at postnatal days (PN) 11, 17, 28, 35, and 42. Iba1 (ionized calcium-binding adapter molecule 1) was the protein selected to study microglial changes. The macroautophagy markers Beclin-1, Atg5, Atg7, microtubule-associated protein light chain 3 (LC3), and lysosomal-associated membrane protein 2 (LAMP2) (involved in chaperone-mediated autophagy (CMA)) were determined. (3) Results: the expression of Iba1 was increased in rd1 retinas compared to the control group at PN17 (after the period of maximum rod death), PN28 (at the beginning of the period of cone death), and PN42. The number of activated (ameboid) microglial cells increased in the early ages of the retinal degeneration and the deactivated forms (branched cells) in more advanced ages. The macroautophagy markers Atg5 at PN11, Atg7 and LC3II at PN17, and Atg7 again at PN28 were decreased in rd1 retinas. At PN35 and PN42, the results reveal alterations in LAMP2A, a marker of CMA in the retina of rd1 mice. (4) Conclusions: we can conclude that during the early phases of retinal degeneration in the rd1 mouse, there is an alteration in microglia and a decrease in the macroautophagy cycle. Subsequently, the CMA is decreased and later on appears activated as a compensatory mechanism.

## 1. Introduction

Retinitis pigmentosa (RP) refers to a group of retinal diseases characterized by the progressive loss of visual abilities due to photoreceptor degeneration. In most cases, rods degenerate first, and then cones die. RP is a rare disease that to date has no treatment [1]. An important characteristic in the pathophysiology of RP is the high variability in terms of the first signs and symptoms of age of onset, the rate of disease progress, and the severity of visual problems. The fact that patients are not able to detect the first signs of the disease makes it very complex to determine the beginning of the disease, with the consequent limitation that this implies in the development of treatment during the first stages of the disease [2]. Because degeneration affects rods first, problems with dark adaptation or some degree of night blindness can appear in an early stage of the disease. This may in turn lead to the progressive loss of the peripheral visual field, with patients developing the so-called “tunnel vision” [3]. Death of photoreceptor cells triggers the migration of retinal pigment epithelium towards the neural retina, leaving pigment deposits in the mid-periphery of the retina [4]. 

Although RP is a hereditary disease, non-genetic biological factors also modulate or contribute to the disease progression. In this sense, inflammatory responses have always played an important role in RP pathogenesis [5,6,7].

The inflammatory responses of glial cells constitute the adaptations to the changes caused by retinal degeneration and the death of photoreceptors and other cells [8]. Glial cells are divided into macroglia and microglia. In the retina, macroglial cells (both astrocytes and Müller cells) are responsible for maintaining the homeostasis of extracellular ions, metabolites, water, and pH, thus controlling the composition of the extracellular fluid [9]. In acute retinal injury, macrogliosis has a neuroprotective role; however, if a chronic lesion occurs in the area, gliosis causes an exacerbation of the problem, increasing vascular permeability and toxins infiltration [10].

In this research work, we are particularly interested in microglia. Microglia are the resident immune cells of the central nervous system (CNS), and the retina is part of it [11]. Microglia cells participate in immune vigilance, neurotrophic assistance, synaptic refinement, and debris removal in the retina [11]. In a healthy retina, microglia can be found in the ganglion cells layer and in the inner and outer plexiform layers [12]. They have a branched shape with long, movable extensions that actively survey the ocular environment [11]. In response to local injury, infection, or retinal degeneration, microglia morphology can change to an amoeboid shape and migrate to the outer retina [11,12]. Glia also play crucial roles at various life stages. During development and maturation, microglia support neuronal survival by interacting with neurons, surveilling the retinal microenvironment with their extensions, regulating synaptic plasticity through phagocytosis, and maintaining synaptic structural integrity and function [13].

Microglia activation has been shown in numerous retinal diseases, including age-related macular degeneration (AMD), glaucoma, diabetic retinopathy (DR), uveitis, retinal detachment, and RP. [11]. In fact, infiltration of damaged retinal regions by microglia has been observed in mice RP models (rd1 and rd10 mice) and in post-mortem samples from RP patients [12].

During the progression of different retinal degenerations, and particularly those that are characterized by progressive photoreceptor death (such as RP), microglia have a dual role, either causing distress or protecting photoreceptors and inner neurons by monitoring, secreting substances, and engulfing cellular debris [13]. Though the exact role of microglia remains uncertain under pathological conditions, it has been demonstrated that an increase in microglial cell activation coincided with the initiation of photoreceptor death in several retinal degeneration animal models [14]. Researchers have demonstrated the proliferation of microglial cells in retinas from P23H-1 rats (carrying a rhodopsin mutation) and Royal College of Surgeons (RCS) rats (with pigment epithelium malfunction), whereas glial fibrillary acid protein (GFAP) over-expression was observed to begin later [14].

Undoubtedly, microglia represent a promising therapeutic target for RD; understanding their precise functions in different pathological contexts is, therefore, needed [13].

Another factor that has also been related to the pathophysiology of RP is autophagy. However, there are still controversies regarding the importance of this process in this disease. Autophagy is a catabolic process that occurs in cells under basal conditions, but can also be induced under stress conditions. Autophagy promotes cytoplasmic material (cargo), including organelles and macromolecules, to be transferred to lysosomes for degradation [15]. This adaptive mechanism is used by the cell to remove old or misfolded proteins, protein aggregates, damaged organelles, and microorganisms, and in addition is an adaptive response that obtains nutrients and energy in situations of cellular stress [16]. When autophagy is excessively induced, cell death is triggered [17]. According to the mechanism by which the cargo (cytoplasmic material) is transferred to the lysosome or vacuole, there are three types of autophagy in mammalian cells: macroautophagy (or simply “autophagy”), autophagy mediated by chaperones (CMA), and microautophagy [18].

Macroautophagy is characterized by the formation of double-membrane vesicles, called autophagosomes, in the cytoplasm. During this process, different cellular components, including organelles, proteins, and other types of cytoplasmic material, are surrounded by phagophores (double-membrane structures that end up expanding and closing to form autophagosomes). Subsequently, the membranes of these structures fuse with that of lysosomes to form autophagolysosomes. Inside the autophagolysosome, degradation of previously sequestered cytosolic components takes place due to the action of hydrolases from the lysosome [19].

In the case of CMA, a chaperone protein attaches itself to the substrate to be degraded. This substrate is a protein and in its amino acid sequence there is a KFERQ pentapeptide (Lys-Phe-Glu-Arg-Gln). The chaperone protein Hsc70 is the only one capable of recognizing the KFERQ sequence to trigger this type of autophagy. Subsequently, the protein substrate binds to the lysosome-associated membrane protein 2A (LAMP2A), and the substrate’s cleavage and multimerization of the receptor take place. Then, the substrate protein and the chaperone are transferred directly into the lysosome for degradation [20].

Microautophagy refers to the direct envelopment of cytoplasmic material by the lysosomal membrane. Microautophagy can be induced by intracellular nitrogen deficiency or by rapamycin. Its main functions are related to the maintenance of the adequate size of cell organelles, the homeostasis of cell membranes, and cell survival under nitrogen-restrictive conditions. Furthermore, microautophagy cooperates with macroautophagy and CMA in a coordinated and complementary way [21].

Autophagy alterations have been related to several retinal pathologies such as AMD, DR, or RP. However, it is not clear whether autophagy plays a protective or harmful role in these pathologies [22]. Even though high levels of autophagy activation exert a protective role in conditions of inflammation and external stress, excessive activation can lead to cell degeneration and deterioration in ocular pathologies [23].

The main objective of this work was to study possible microglia and autophagy alterations in one RP animal model at different stages of the disease (when rods are dying and later, when there are almost no rods, and cones are the cells that die) to determine which of these changes occur earlier and if they are possible target processes in the search for new therapies. Iba1 (ionized calcium-binding adapter molecule 1) was the protein selected to study microglial changes. Iba1 constitutes one of the most suitable markers for the structural analysis of microglial cells, both under normal conditions and in the presence of pathological alterations in nervous tissue [23]. Herein, we have also determined the possible alterations of different markers of macroautophagy, such as protein Beclin-1, proteins related to autophagy, Atg5 and Atg7, and finally the microtubule-associated protein light chain 3 (LC3). We have also studied the expression of lysosomal-associated membrane protein 2 (LAMP2) as an approach to recognize changes in autophagy mediated by chaperones.

The animal model used in this study was rd1 mice. The mutation in rd1 mice results in severe and early retinal degeneration, which is due to the insertion of a murine virus that introduces a nonsense mutation in exon 7 of the gene that encodes for phosphodiesterase 6β (PDE6β) [24,25]. The mutation of PDE6β is also found in humans and has been related to the cases of autosomal recessive RP which constitute approximately 5% of the cases of RP in humans [26]. Two characteristics resulting from the mutation have been described: a deficiency in the catalytic activity of PDE6β and a subsequent accumulation of cyclic guanosine monophosphate (cGMP) [27]. The degenerative process in this animal model starts very early, even before the retina becomes mature, around post-natal (PN) day 10 [28]. The peak of rod photoreceptor cell death occurs between PN11 and 14 [29]. By PN30, the mutation-dependent degeneration is almost completed, with one or two rows of photoreceptors remaining, mainly cones, which will eventually die within the next 6 months [30].

## 2. Materials and Methods

For this study, male and female mice were used. C3H, or control mice, belonged to the strain C3Sn.BLiA-Pde6b+/DnJ, homozygous for Pde6b+. This C3H congenic strain lacks the retinal degeneration gene Pde6brd1 that is characteristic of C3H substrains, and thus offers most of the strain characteristics of the C3H background, but without the early-onset retinal degeneration. Mice from the rd1 mice (C3H/HeJ) model of retinal degeneration were also used; these mice are homozygous for the retinal degeneration 1 mutation (Pde6brd1), causing blindness by weaning age. All mice were derived from the Jackson Laboratory colony (The Jackson Labs, Bar Harbor, ME, USA). Four mice per group were used for immunohistochemistry determinations and four mice per group were used for Western blot techniques.

The Animal Ethics Committee of the CEU Cardenal Herrera University approved the animal care and protocols (reference 11/013), which adhered to the ARVO Statement for the Use of Animals in Ophthalmic and Vision Research.

Mice had unrestricted access to water and food (Harlan Ibérica S.L. (Barcelona, Spain)) and were housed in controlled conditions of humidity (60%), temperature (20 °C), and light/dark cycles (12 h). Finally, mice were euthanized at postnatal days 11, 17, 28, 35, and 42.

The immunohistochemistry technique was chosen to determine the expression of Iba1, as it allows us to identify the location of the Iba1-positive cell somas, their morphology, and their degree of activation. Eyes were enucleated. Each eye’s cornea was pierced with a needle, and then the eye was immediately immersed in 4% paraformaldehyde (PFA) for 2 h. Subsequently, eyes were washed three times for 10 min with 0.1 M phosphate-buffered saline (PBS). The eyes were cryoprotected using progressively increasing sucrose–PBS solutions (10–30%) and embedded in Tissue Tek (Sakura Europe, Barcelona, Spain).

Retinal sections of 8 µm were obtained using a cryostat (Leica CM 1850 UV Ag protect, Leica Microsistemas S.L.U., Barcelona, Spain) and placed on superfrost slides (Thermo Fisher Scientific, Braunschweig, Germany), which were kept at −20 °C. The tissue sections were rehydrated with PBS for 15 min, and then blocked with 5% normal goat serum, PBS-BSA 1%, and Triton 0.3% for 1 h at 4 °C to prevent nonspecific binding. Afterward, the sections were rinsed three times for 10 min with PBS-Triton 0.3%, and the primary antibody (listed in Table 1) was incubated with PBS-Triton 0.3% overnight at 4 °C. Following three rinses of 10 min each with PBS-Triton 0.3%, the secondary antibody, Alexa 488 (Invitrogen, Life Technologies, Madrid, Spain), was incubated for 1 h at 4 °C. Finally, the sections were mounted using Vectashield mounting medium with DAPI (Vector, Burlingame, CA, USA). Fluorescence microscopy was performed using a Nikon DS-Fi1 camera attached to a Leica DM 2000 microscope. The Leica Application Suite version 2.7.0 R1 software (Leica Microsystems SLU, Barcelona, Spain) was used for analysis. A negative control (with primary antibody omission and only use secondary antibody) has also been performed and it showed the absence of Iba1 detection (Appendix A). Iba1 expression was assessed by quantifying the number of positive cells similarly to Di Pierdomenico et al. [14]. Within each animal, three sagittal cross sections containing the optic disk were chosen based on section quality. Three photographs (at 20× magnification) were captured for each section. The distance between the optic disc and the retinal periphery was measured and the three images were taken in each section at a distance equivalent to 50% of the length between the optic disk and the retinal periphery. The numbers of the different microglial (branched and ameboid) cells were subsequently counted in each photomicrograph according to their morphology. Image quantification, covering an approximate length of 500 μm, was conducted manually with the assistance of ImageJ software. These individual counts were pooled to calculate the average number of microglial cells per animal (four animals per age were analyzed, n = 4).

To analyze autophagy markers, pools of two fresh retinas were frozen at −80 °C until their use. Retinas were homogenized using radioimmunoprecipitation (RIPA) buffer, which consisted of 150 mM NaCl, 1% Nonidet P-40, 0.5% sodium deoxycholate, 0.1% sodium dodecyl sulfate (SDS), and 50 mM Tris at pH 8. The Bradford method was used to determine the protein concentration [31]. Protein samples (75 μg) were run during 2 h on 10–15% acrylamide: bisacrylamide gels. The proteins were then transferred onto nitrocellulose membranes (GE Healthcare Life Sciences, Barcelona, Spain) and blocked for 1 h with 0.01 M PBS-Tween 20 (0.1%) containing 5% (*w*/*v*) non-fat milk.

The membranes were probed with antibodies against Beclin-1 (Santa Cruz, Santa Cruz, USA), Atg5 (Sigma-Aldrich, Spain), Atg7 and LC3 (both from Cell Signaling, MA, USA), and LAMP-2A (Invitrogen, CA, USA) as listed in Table 1. The membranes were incubated overnight at 4 °C, and the bound antibodies were visualized using a horseradish peroxidase-coupled secondary antibody (F(ab′)2-HRP, goat anti-rabbit) (Santa Cruz Biotechnology, Santa Cruz, USA) (Table 2). The signal was detected using an enhanced chemiluminescence (ECL) developing kit (Amersham Biosciences, Buckinghamshire, UK), and the blots were quantified using densitometry (ImageQuant™TL, GE Healthcare Life Sciences, Barcelona, Spain). The original membranes of the Western blots can be found in Appendix A.

The Wako anti-Iba-1 antibody was chosen for immunohistochemistry due to its proven specificity in the mouse retina, as demonstrated in the study by Zhang et al. [32]. Similarly, the antibodies used in this study to determine the autophagy status in the retina via Western blot were also confirmed in the literature to be specific for their use in mouse retinas (Anti-Beclin-1 [33], Anti-Atg5 [34], Anti-Atg7 [35], Anti-LC3 [36] and Anti-LAMP-2A [37]).

The results are presented as mean values ± standard deviations from four animals in each group. Prior to conducting any test, we verified that the populations followed a normal distribution and that there was homogeneity of variances.

To study Iba1 expression, a two-way analysis of variance (ANOVA) was conducted, with age as the primary treatment and strain selected as the second factor. In cases where the ANOVA revealed a significant difference, a post hoc Dunnett’s T3 test was employed.

Between the control and rd1 groups within each age, a Student’s t-test was used to compare the results obtained for autophagy marker expression. Autophagy marker differences along ages were not studied because the Western blots of the different ages were not performed on the same membrane.

A 95% confidence interval was established to assess significance, considering data with a *p*-value less than 0.05 (*p* < 0.05) as significant. Statistical analysis of the data was performed using Microsoft Excel 2016 and RCommander 3.4.1.

## 3. Results

To characterize changes in microglia and autophagy markers in our mouse model with retinitis pigmentosa (rd1) in comparison to control mice over time, we analyzed retinas from PN 11, 17, 28, 35, and 42 for the following reasons: (i) PN11 marks the initiation of a time period when the first wave of cell death occurs, primarily affecting the rods [29]; (ii) at PN17, the cell death process initiated on day 11 is still ongoing until PN20; however, by this age, the outer nuclear layer (ONL) has already completely degenerated and, therefore, day 17 was chosen to examine the retina prior to the near-total degeneration of the ONL on postnatal day 20; (iii) the second wave of cell death, which involves cones, occurs from postnatal day 28 onwards [29]; to study the cone death period beyond 28 days, we extended our investigation at 7-day intervals, resulting in postnatal ages 35 and 42 as the most suitable time points to examine cone cell death.

### 3.1. Microglia Expression in the Retina of the Rd1 Mice Model

Iba1 is a microglia/macrophage-specific marker that has been extensively employed for microglial detection. In the present study, Iba1 expression was examined by analyzing retinal images of C3H (control) and rd1 (RP model) mice at PN days 11, 17, 28, 35, and 42 (Figure 1). Iba1-positive cells were observed in the ganglion cell layer (GCL), inner plexiform layer (IPL), inner nuclear layer (INL), outer plexiform layer (OPL), and outer nuclear layer (ONL) of the rd1 retina.

When quantifying Iba1-positive cells, the number of Iba1-labeled cells per arbitrary unit area in the central retina of the animals was determined. However, it was considered necessary to differentiate cell morphology. Consequently, total positive Iba1 cells, as well as Iba1 cells with an amoeboid shape (indicative of activated microglia) and cells with a branched shape (representing resting microglia) were quantified (Figure 2A–C). This differentiation allowed us to determine not only the presence of microglia cells but also their activity.

At postnatal day 11 (PN11), the control group had an average of 0.4 total Iba1-positive cells per arbitrary unit area, while the rd1 group exhibited a lower amount (0.25 cells per unit area), although this difference was not statistically significant. However, at postnatal days 17 and 28 (PN17 and PN28), the control group had 0 and 0.03 Iba1-labeled cells per unit area, respectively. In contrast, the rd1 group showed an increase in this parameter at these two postnatal ages, reaching values of 0.83 and 0.60 cells per unit area. At postnatal day 35 (PN35), there were no statistically significant differences in the number of labeled cells per unit area between the control group and the rd1 group. However, there was a statistically significant difference in the number of labeled cells per unit area in the central retina between the control group (0.87) and the rd1 group (1.72) (*p* = 0.0044) on postnatal day 42 (PN42) (Figure 2A).

Regarding the temporal evolution of the resting form (branched form) of microglia, an increase in the number of these cells was observed between PN11 and PN17 in the rd1 mice, reaching a value of 0.35 cells per unit area labeled with Iba1, which was maintained at PN28 (0.32 cells per unit area). In the PN28-PN35 interval time, an increase in branched cell density was observed in both control and rd1 retinas. In the subsequent interval period studied (PN35-PN42), the control retinas exhibited a decrease in the number of branched cells, while the rd1 group showed an increase to 1.72, establishing a statistically significant difference at the final age studied (PN42) (*p* = 0.003). We can conclude that the number of branched positive Iba1 cells increased with the age and progression of retinal degeneration in the rd1 mice (Figure 2B).

A significant increase in the number of ameboid Iba1 cells (indicating the active form) was observed in the rd1 retinas compared to the control ones at PN17. Subsequently, in the rd1 group, there was a time-dependent decrease in labeling from PN17 to PN42. On the other hand, the retinas from the C3H (control) group exhibited a decrease in amoeboid-shaped cells between PN11 and PN17, with the density of these cells remaining close to zero over time (Figure 2C).

### 3.2. Quantification of Macroautophagy and CMA Markers in the Retinas of Control and Rd1 Mice at Different Postnatal Ages

The results regarding Iba1 expression may point to the fact that there are microglial alterations and inflammatory processes triggered in the retina during RP degenerations that may alter cellular homeostasis. Under these circumstances, another biochemical process also could be altered, such as autophagy. Consequently, we have also determined the autophagy markers Beclin-1, Atg5, Atg7, and LC3 (participants in the macroautophagy process), and LAMP2A (involved in chaperone-mediated autophagy) using the Western blot technique to assess the role of this biochemical process in the evolution of RP.

Macroautophagy is characterized by the formation of autophagosomes. One of the initial steps in the assembly of autophagosomes is the recruitment and activation of the class III phosphatidylinositol 3-kinase complex consisting of Beclin-1, VPS34, VPS15, and ATG14 proteins [38]. For this reason, the first autophagy marker we studied in the rd1 and control mice retinas was Beclin-1.

At the earliest postnatal ages examined (PN11 and PN17), the optical densities of the control group samples for Beclin-1 labeling were comparable to those of the rd1 group samples. At PN28, Beclin-1 expression showed a slight decrease in the rd1 group compared to the control group, but without reaching statistical significance (Beclin-1: rd1 group optical density 0.96 vs. control group 1.32). No differences were observed in Beclin-1 expression between control and rd1 retinas at postnatal ages 35 and 42 (Figure 3).

After the formation of the phagophore and the important role of Beclin-1, the ATG12 and LC3 conjugation systems are key in mediating the expansion and nucleation of the phagophore into an autophagosome [39]. Atg7 participates both in the binding of Atg5 to Atg12, and in the reaction prior to LC3 lipidation [39]. Therefore, we examined Atg5, Atg7, and the ratio LC3II/LC3I; these three markers participate in the same phase of the macroautophagy process: elongation of the autophagosome and fusion with the lysosome.

At postnatal day 11, we only observed a statistically significant decrease in Atg5 expression in the rd1 retinas compared to the control group, and no differences between control and RP retinas were observed in the other markers of the macroautophagy process (Atg7 and LC3II/LC3I) (Figure 4, Figure 5 and Figure 6).

Contrary to what occurs at PN11, no statistically significant changes were observed between the control group and rd1 in Atg5 expression at postnatal day 17 (PN17) (Ac/APx ratio 0.89 for the control group vs. 0.79 for the rd1 group (Figure 4).) In contrast, Atg7 was significantly decreased in rd1 mice compared to the control group (*p* < 0.05) (Figure 5). The LC3II/LC3I ratio is also significantly lower in the rd1 animals compared to the control ones at PN17 (*p* < 0.05) (Figure 6).

At postnatal age 28 (PN28) we observed a statistically significant decrease in Atg7 in the rd1 retinas compared to the control group (*p* < 0.05) (Figure 5). Regarding LC3II/LC3I, no significant differences were found between the rd1 and the control mice at PN28 (Figure 6).

At postnatal days 35 and 42, Atg5 and Atg7 present similar values of optical density in the rd1 and control groups (Figure 5 and Figure 6). The LC3II/LC3I ratio shows very similar values in both control and rd1 retinas at PN days 35 and 42 (Figure 6).

Finally, the CMA autophagy marker, LAMP2A, was studied in control and rd1 retinas of mice with 11, 17, 28, 35, and 42 postnatal ages. No differences were observed in the expression of LAMP2A in the retinas of control and retinitis pigmentosa mice at PN days 11, 17, and 28. At 35 days of age (PN35), LAMP2A presents statistically significant differences between control and rd1 retinas. These differences are due to a decrease in the LAMP2A protein in the rd1 animals compared to the control ones (Figure 7). Finally, and as occurs with PN 42, the LAMP2A marker presents statistically significant differences between both studied group animals; however, at this age LAMP2A is increased in the rd1 retinas compared to the control group (optical density difference 0.14) (Figure 7).

## 4. Discussion

In the first part of this work, we have examined the possible alterations of microglia cells in retinitis pigmentosa, as well as the sequence of these changes as the retinal degeneration progresses. In this sense, we have observed that the expression of the Iba1 microglia marker was increased in the retinas of the rd1 mice (an animal model of RP) compared to the control group at PN17 (after the period of maximum death of the rods), PN28 (at the beginning of the period of death of the cones), and PN42. However, at PN11, when the retina is not yet fully formed, no significant differences appeared between both animal groups. In addition, the activated (ameboid cells) and deactivated forms (branched cells) of the cells marked with Iba1 were quantified. The number of ameboid cells increased in the early ages of the study and the second form in more advanced ages, when the period of maximum death of the cones had passed.

Other studies have examined the evolution of the concentration of microglia in the retinas of the rd1 animal model. According to Sancho-Pelluz et al. [40], the central retina of the rd1 mouse presents cells with CD11b labeling, a microglia marker, and the expression of this protein increases between PN12 and PN21, a result consistent with the increase in Iba1 labeling between postnatal days 11 and 17, shown in Figure 2 of this work.

We have also found an evident increase in marked microglia, both with an amoeboid and branched shape, between PN11 and PN17. This increase, particularly in Iba1 cells of amoeboid morphology, was also shown by Zhou et al. in 2018, at PN14 and in the rd1 model. Furthermore, they demonstrated the protective effect of alpha-1-antitrypsin on neuroinflammation and degeneration at the retinal level in rd1 mice [41]. These same authors published, a year earlier, a study on the evolution of microglia, in terms of number and morphology, in the rd1 retina at postnatal ages PN14, PN21, PN28, and PN180 [42]. These authors identified the peak density of microglia with amoeboid majority at PN14, with decreases at later ages. These results are similar to the ones obtained in this study; we have not studied PN14, but we have observed no changes at PN11 and an increase in the total number of microglial cells, and particularly ameboid Iba1-positive cells, at PN17. At PN180, the microglia had a branched shape; that is, after the process of degeneration and death of photoreceptors, the microglia returned to their inactive state. This is again consistent with our results, which show this predominance of the branched form (inactive) at PN35 and PN42.

All studies related to microglia in rd1 retinas mentioned so far establish in one way or another a close relationship between microglia and the degenerative process of photoreceptors, which takes place in the ONL of the RP retina. This has also been demonstrated in other non-murine animal models that have described, in rat models, the activation and migration of microglia towards the ONL, from the beginning of the degeneration of photoreceptor cells [43]. Our results are also in agreement with those obtained by He et al. [44], who used the Royal College of Surgeons (RCS) rat model characterized by a recessively inherited mutation in the receptor tyrosine kinase gene Mertk, resulting in secondary degeneration of photoreceptors between postnatal days 20 and 60. Although, in our rd1 mouse model, the retinal degeneration process is faster, the trend of increased Iba-1 cells in the retina is similar in comparison with their respective control group.

Gupta et al. carried out immunohistochemical techniques for the detection and postmortem analysis of microglia in healthy human retinas, as well as in others with RP, age-related macular degeneration (AMD), and late-onset retinal degeneration [45]. Their work showed how microglia had a star-shaped shape in normal retinas, as well as being small in size and associated with blood vessels in the inner area of the retina; however, in the pathological retinas, numerous activated microglia cells were observed in the ONL, in the regions where cell death was occurring. In the latter case, the cells were larger in size and amoeboid in shape.

Growing evidence has demonstrated that autophagy proteins are involved in the physiology and pathology of the retina and that autophagy alterations may contribute to retinal degeneration [22,46]. In the present study, an analysis of the autophagy markers in the retinas of control and RP mice was carried out.

Autophagy has been related to different brain neurodegenerative diseases. Structures involved in the autophagy process, such as autophagosomes, accumulate in the brains of patients with neurodegenerative pathologies such as Alzheimer’s disease, spongiform encephalopathies, or Parkinson’s or Huntington’s disease [47,48]. Alterations or defects in the autophagy cycle have also been observed in eye pathologies such as glaucoma, cataracts, AMD, diabetic retinopathy, eye tumors, and infections [23].

Wang et al. showed the involvement of autophagy in cell degeneration in the retinas of Drosophila melanogaster [49]. In their study, the inhibition of TOR (mediator that negatively regulates autophagy) or the genetic induction of autophagy suppressed cell death in retinal degeneration. However, even though high levels of autophagy activation exert a protective role under conditions of inflammation and external stress, excessive activation can lead to degeneration and cell deterioration in ocular pathologies [23].

The involvement of the autophagy process in RP in rd1 mice is also evident in the present work: in each one of the five ages studied (PN11, PN17, PN28, PN35, and PN42), there are statistically significant differences revealed in at least in one autophagy marker of the five studied, between the rd1 retinas and the control group. In this regard, we have demonstrated that the macroautophagy markers Atg5 at PN11, Atg7 and LC3II at PN17, and Atg7 again at PN28 are decreased in rd1 retinas (Figure 4, Figure 5 and Figure 6).

Decreases in retinal Atg5 have previously been studied in the retina. The deletion of the gene that encodes for Atg5 from the rod photoreceptors of the retina results in the accumulation of the phototransduction protein transducin and the degeneration of these neurons [50]. It has also been demonstrated that the genetic material responsible for apoptosis and responsible for autophagy are co-expressed in photoreceptors that develop death and degeneration [51]. These results suggest that autophagy participates in photoreceptor cell death, possibly by initiating apoptosis [51]. In this regard, we have determined that Atg5 is already decreased at PN11 in rd1 retinas, when the development of the retina has not even been completed in this animal model (Figure 4).

However, other studies have observed an increase in the values of autophagy markers (in this case, lysozyme and cathepsin S in autophagosomes and autophagolysosomes) in rd1 mice [52]. This increase took place on days 14–15 postnatally, a period in which the peak of both cell death in general (TUNEL), neuroinflammation (GFAP), and apoptosis (caspase 1) also occurred [52].

In rd1 mice, rod death occurs during the second and third postnatal week, and the death of the cones takes place after the fifth week and continues for several months [29]. Thus, at approximately PN28, it can be said that the degeneration of the cones begins. At this postnatal age, our data showed that the macroautophagy marker Atg7 is significantly decreased in the rd1 group, compared to the control group (n = 4 in each group) (Figure 6). These preliminary results may confirm others that affirm that the macroautophagy process is detected only for 24 h at the beginning of the degeneration of the cones in rd1 mice, prior to chaperone-mediated autophagy (CMA) [53].

CMA is altered later in rd1 mice, at the ages at which cone degeneration occurs [53]. This agrees with the preliminary results of this study; at the two later ages studied, PN35 and PN42, the results reveal statistically significant alterations in LAMP2A, a marker of chaperone-mediated autophagy (CMA) in the retinas of rd1 mice. However, we have observed that LAMP2A is decreased at PN35 and increased at PN42 (Figure 7). The increase in CMA observed in the rd1 retina may be a compensatory process. CMA activation has been described in vitro in macroautophagy-deficient cell lines [54], as a compensatory process. Specifically in the retina, a decrease in the Atg5 marker has been classified as the cause of a compensatory mechanism in the form of CMA activation [55].

Therefore, we can preliminarily conclude that during the early phases of life and retinal degeneration (between PN11 and PN28), in the retina of the rd1 animal model, there is an alteration in microglia and the macroautophagy cycle. Subsequently, the CMA, as stated because of LAMP2A expression, is decreased and later appears activated as a compensatory mechanism (Figure 8). The results obtained in this study are limited because the number of animals in each group was four, and new studies should be performed to confirm them.

## 5. Conclusions

Based on the above, it would be reasonable to conclude that inflammation and macroautophagy processes could be possible alternatives in the treatment of degenerative pathologies of the retina in the initial stages of RP. In this phase, cones, which are mainly responsible for human vision, have not yet degenerated, thus allowing a very high quality of life for patients if retinal degeneration can be stopped or slowed down in this phase. On the other hand, chaperone-mediated autophagy (CMA) would constitute a possible therapeutic target when cones are degenerating.

## Figures and Tables

**Figure 1 biomolecules-13-01277-f001:**
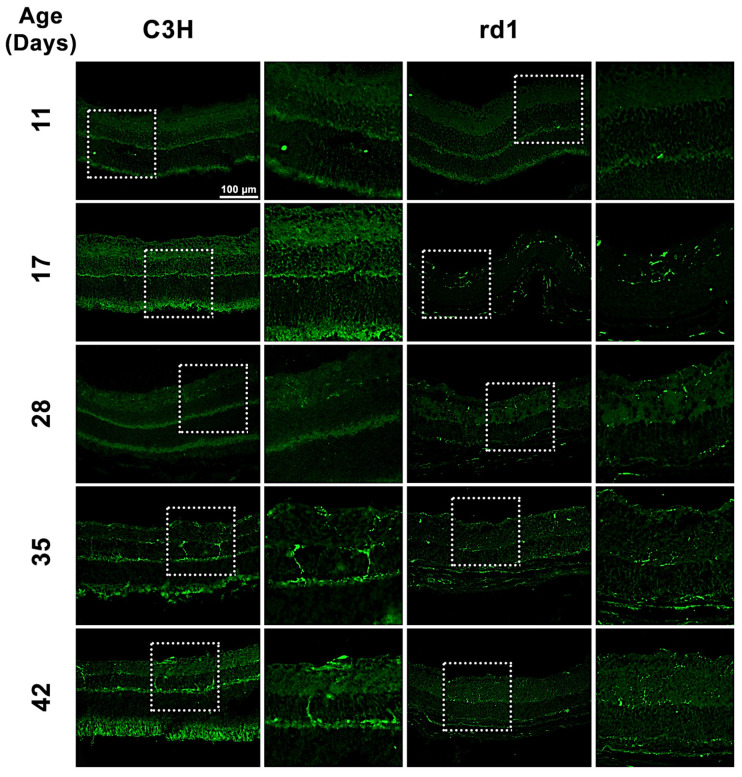
Microglia expression in control and rd1 retinas. Immunohistochemical images of retinal sections that underwent immunostaining with the Iba1 antibody at PN11, PN17, PN28, PN35, and PN42 (magnification: 40×). The images on the left correspond to C3H (control) mice, while the images on the right part represent retinas from rd1 mice. Iba1-positive cells were detected in GCL, IPL, INL, OPL, and ONL of the retina.

**Figure 2 biomolecules-13-01277-f002:**
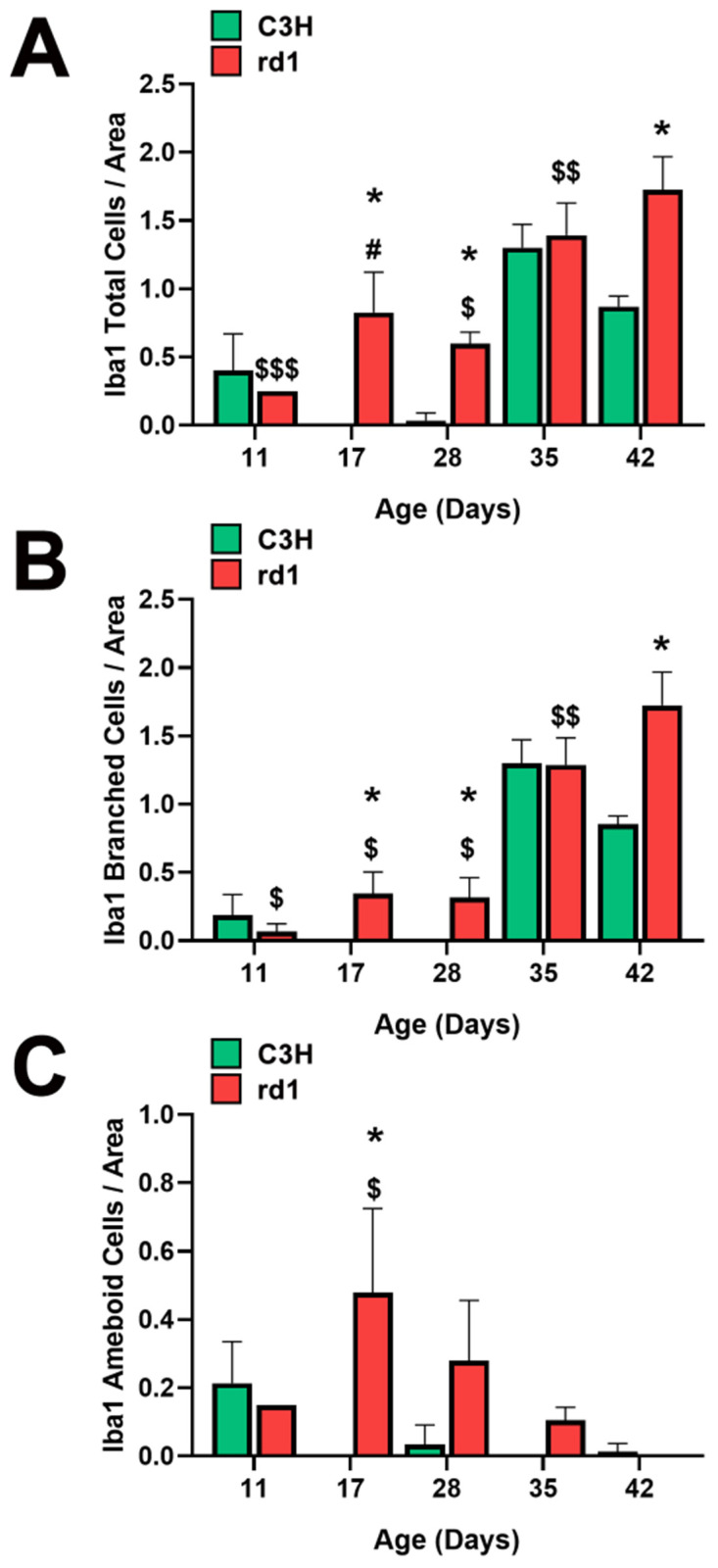
Microglia morphologic analysis. (**A**) Number of total Iba1-positive cells in the retinas of control and rd1 mice at PN11, PN17, PN28, PN35, and PN42. (**B**) Number of branched Iba1-positive cells in the retinas of control and rd1 mice at PN11, PN17, PN28, PN35, and PN42. (**C**) Number of ameboid Iba1-positive cells in the retinas of control and rd1 mice at PN11, PN17, PN21, PN28, PN35, and PN42. (* *p* < 0.01 vs. control at the same postnatal age; $ *p* < 0.05 vs. PN35 and PN42; $$ *p* < 0.05 vs. PN42; $$$ *p* < 0.05 vs. PN17, PN35 and PN42; # *p* < 0.05 vs. PN28, PN35 and PN42, according to the two-way analysis of variance). The graphs represent the number of cells labeled with Iba1 per arbitrary unit of area in the central retinas of both groups (arbitrary units of area = cells/μm^2^ × 10^3^). Four mice per group were used and three histological sections of one retina of each mouse were quantified.

**Figure 3 biomolecules-13-01277-f003:**
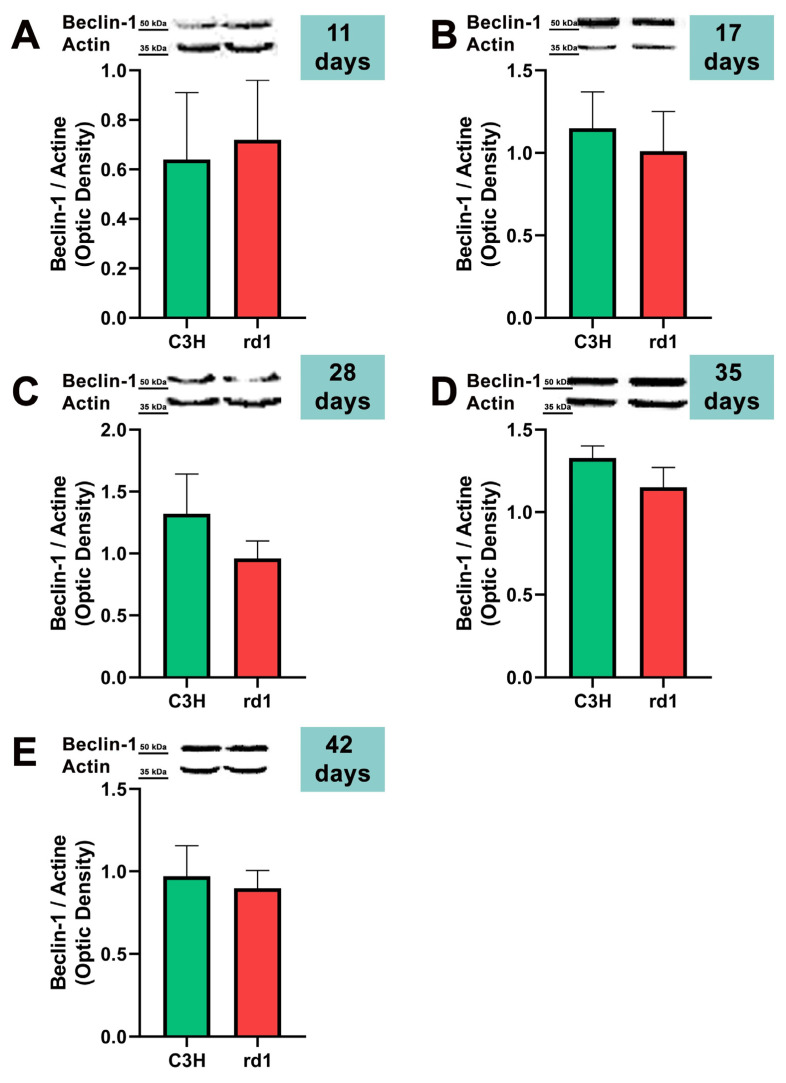
Expression of the autophagy marker Beclin-1 in control and rd1 retinas. (**A**–**E**) Representative Western blots (Image Quant™TL photos) and corresponding densitometrical analyses of Beclin-1 in retinas from control mice compared to rd1 animals at PN11, 17, 28, 35, and 42. Data from at least 4 animals per group are presented as mean ± standard deviation. Actin was used as loading control; optic density is expressed by means of arbitrary units.

**Figure 4 biomolecules-13-01277-f004:**
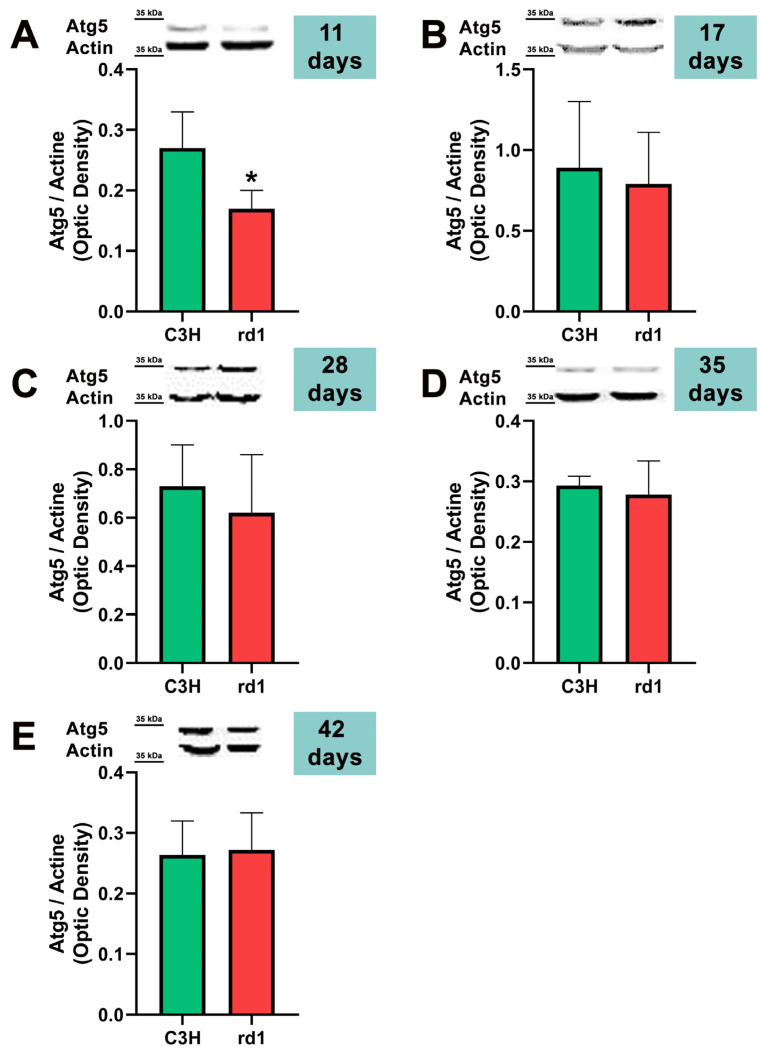
Expression of the autophagy marker Atg5 in control and rd1 retinas. (**A**–**E**) Representative Western blots (Image Quant™TL photos) and corresponding densitometrical analyses of Atg5 in retinas from control mice compared to rd1 animals at PN11, 17, 28, 35, and 42. Data from at least 4 animals per group are presented as mean ± standard deviation. Actin was used as loading control; optic density is expressed by means of arbitrary units. (* *p* < 0.05 vs. C3H retinas at the same postnatal age).

**Figure 5 biomolecules-13-01277-f005:**
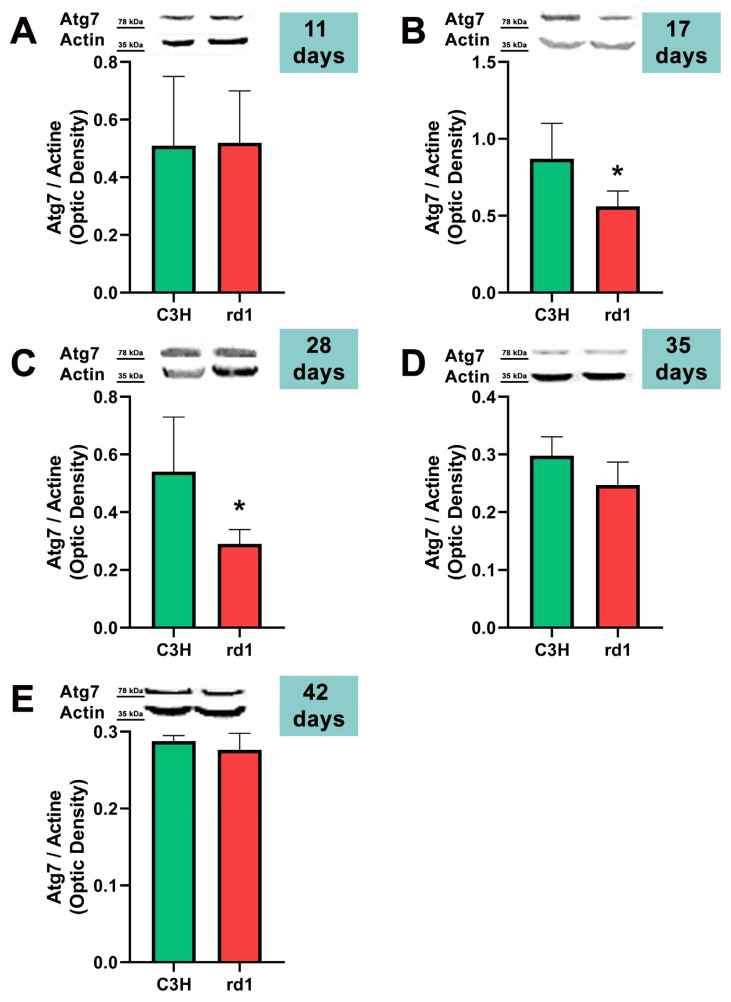
Expression of the autophagy marker Atg7 in control and rd1 retinas. (**A**–**E**) Representative Western blots (Image Quant™TL photos) and corresponding densitometrical analyses of Atg5 in retinas from control mice compared to rd1 animals at PN11, 17, 28, 35, and 42. Data from at least 4 animals per group are presented as mean ± standard deviation. Actin was used as loading control; optic density is expressed by means of arbitrary units. (* *p* < 0.05 vs. C3H retinas at the same postnatal age).

**Figure 6 biomolecules-13-01277-f006:**
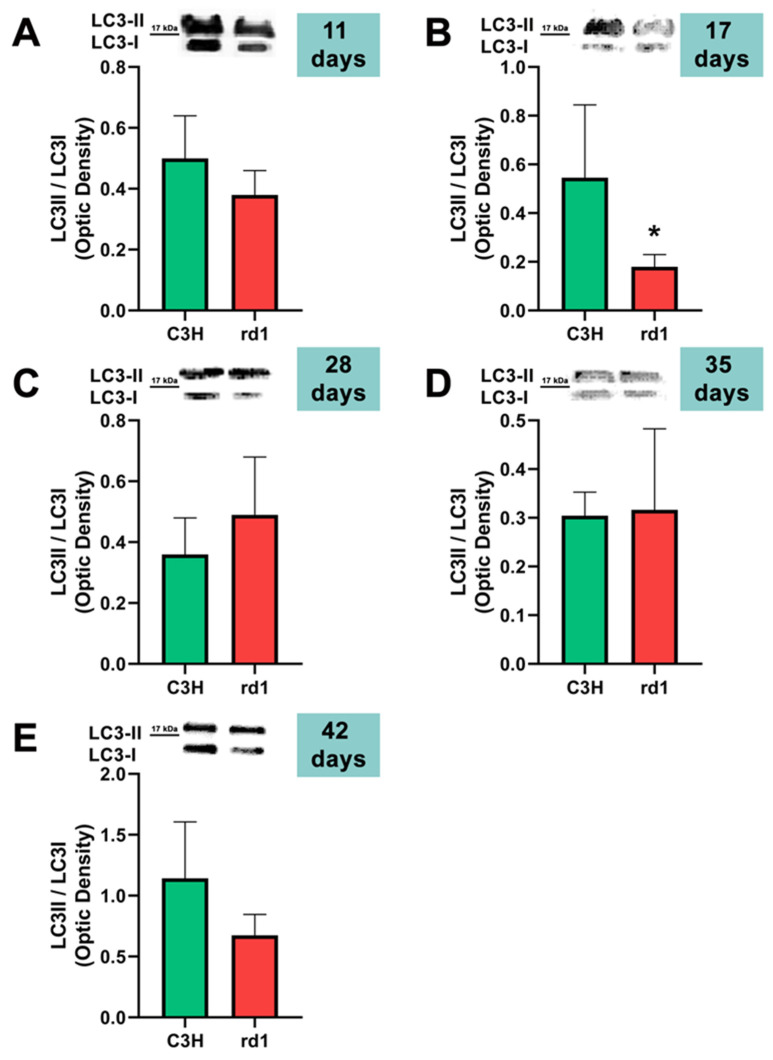
LC3II/LC3I ratio in control and rd1 retinas. (**A**–**E**) Representative Western blots (Image Quant™TL photos) and LC3II/LC3I ratios in retinas from control mice compared to rd1 animals at PN11, 17, 28, 35, and 42. Data from at least 4 animals per group are presented as mean ± standard deviation. (* *p* < 0.05 vs. C3H retinas at the same postnatal age).

**Figure 7 biomolecules-13-01277-f007:**
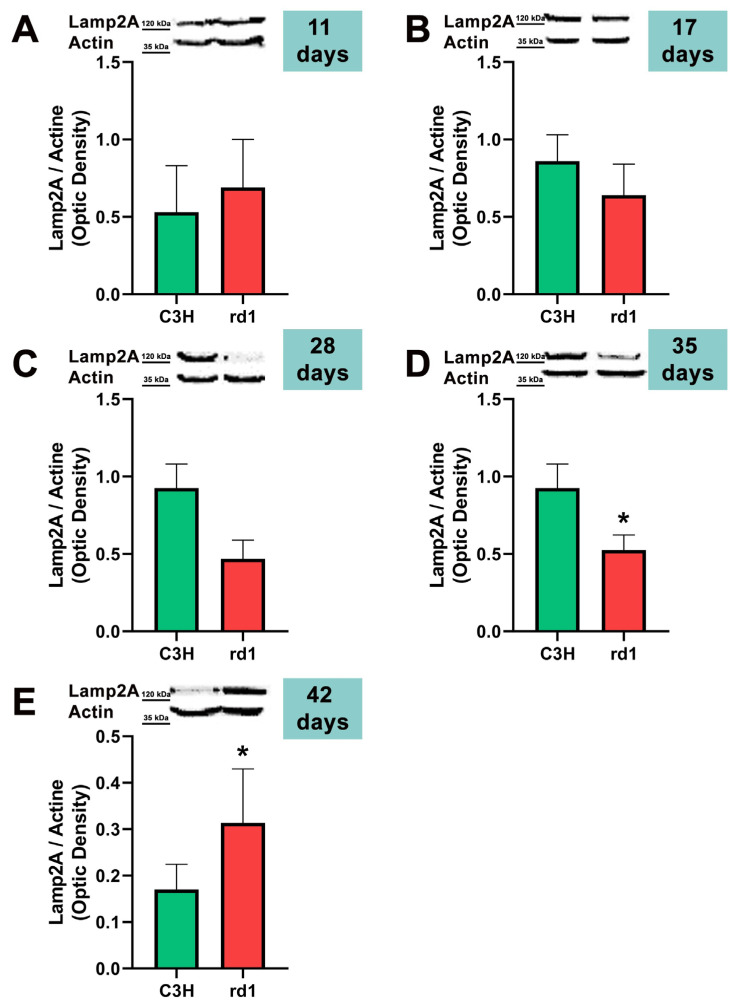
Expression of the chaperone-mediated autophagy marker (CMA) LAMP2A in control and rd1 retinas. (**A**–**E**) Representative Western blots (Image Quant™TL photos) and corresponding densitometrical analyses of LAMP2A in retinas from control mice compared to rd1 animals at PN11, 17, 28, 35, and 42. Data from at least 4 animals per group are presented as mean ± standard deviation. Actin was used as loading control; optic density is expressed by means of arbitrary units. (* *p* < 0.05 vs. C3H retinas at the same postnatal age).

**Figure 8 biomolecules-13-01277-f008:**
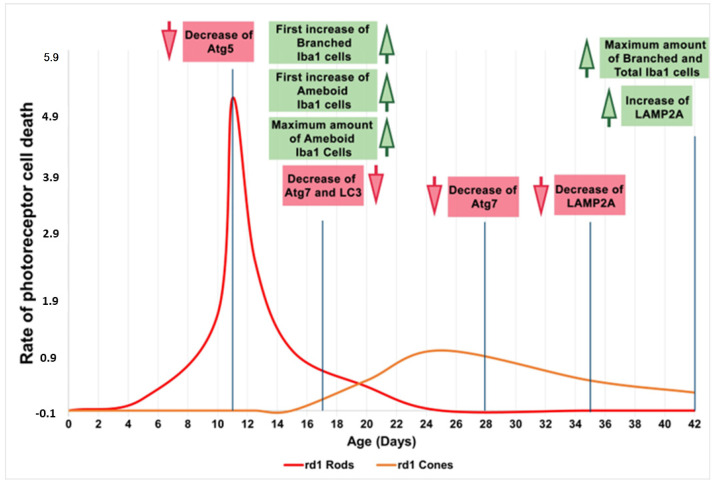
Microglia and autophagy alterations in the retinas of rd1 mice compared to the rate of rod and cone degeneration. First, microglia alterations occur at postnatal age 17 (PN17). Atg5 at PN11, Atg7 and LC3II at PN17, and Atg7 again at PN28 are decreased in rd1 retinas. LAMP2A expression is decreased at PN35 and later increased at PN42. Red arrows indicate decrease and green arrows indicate increase (Note: the curves depicting the progression of rod and cone cell death in the rd1 mouse have been created for informational purposes. We did not conduct a specific experiment for this study; rather, they have been manually drawn based on our prior experience and data that have been previously published).

**Table 1 biomolecules-13-01277-t001:** Characteristics and working concentrations of the primary antibodies used for immunohistochemistry and Western blot techniques.

Antibody	Description	Concentration	Predicted Molecular Weight	Company	Catalog Number
Anti-Iba1	Monoclonal rabbit	1:500	17 kDa	Wako	Z0334
Anti-Beclin-1 (H-300)	Polyclonal rabbit	1:1000	50 kDa	Santa Cruz Biotechnology	sc-11427
Anti-Atg5	Polyclonal rabbit	1:500	35 kDa	Novus Biologicals	NB110-53818
Anti-Atg7	Polyclonal rabbit	1:1000	78 kDa	Cell Signaling	2631S
Anti-LC3	Polyclonal rabbit	1:1000	17 kDa	Cell Signaling	2775S
Anti-LAMP-2A	Polyclonal rabbit	1:1000	120 kDa	Invitrogen	51-2200
Anti-β-Actin-Peroxidase	Monoclonal rabbit	1:30,000	35 kDa	Sigma-Aldrich	A3854

**Table 2 biomolecules-13-01277-t002:** Characteristics and working concentrations of the secondary antibodies used for immunohistochemistry and Western blot techniques.

Antibody	Description	Concentration	Company	Catalog Number
Alexa Fluor^®^ 488	Goat anti-rabbit	1:200	Life Technologies	A11034
IGG F(AB′)2-HRP	F(ab′)2 goat	1:5000	Santa Cruz Biotechnology	sc-3837

## Data Availability

Not applicable.

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
