# Peer review of "Sequences of Alterations in Inflammation and Autophagy Processes in Rd1 Mice"

_biomolecules, 2023, doi:10.3390/biom13091277_

Round 1
Reviewer 1 Report
In this manuscript, the authors study the microglial activation, morphology, migration, and autophagy markers in the retinas of rd1 mice in the early stage of the degeneration.
The manuscript is well written, and the topic and the results are very interesting.
However, there are some important concerns that the authors should improve.
Introduction:
• Considering that the microglia cells are an important part of the results, the authors should explain whit more detail the role of microglia in the early stage of retinitis pigmentosa (RP). Different authors have demonstrated that in RP animal model the microglial cell activation starts earlier that photoreceptor death ( doi: 10.3389/fnana.2017.00014; DOI: 10.1038/srep33356).
Material and methods:
• The authors should explain in more detail the control and experimental animal groups. To make it easier to understand the manuscripts for the readers the authors should indicate the n for each control and experimental group and explain better the control group.
• The authors should describe in more detail how the microglial cells were quantified whether manually or automatically.? Specify in which areas of the retina the microglia cells have been quantified and whether they have always been the same in all the slices analyzed, because it has been shown that in animal models of RP microglia activation is not the same in the dorsal part of the retina as in the ventral part (DOI: 10.3389/fnana.2017.00014, doi: 10.4103/1673-5374.251204).
• Authors should specify the size/portion of the retinal that has been quantified (define arbitrary unit).
• Authors should specify how the branching and morphology of microglia cells have been quantified.
Results:
• Figure 1 and results of iba1+ cell quantifications:
The iba-1 labelling shown in Figure 1 is very poor and does not allow a real quantification of the microglia cells present in the retina. Other authors, some also cited in this article (Zohu et al. 2017 DOI: 10.3389/fnana.2017.00077), show a significantly higher number of microglia cells in the retina of rd1 mice. For this reason, my suggestion is to double-check the quantifications of iba-1+ cells with images where the iba-1 labelling is cleaner, allowing the identification of all iba1 cells present in the retina.
Discussion.
It would be interesting if the authors included a paragraph in the discussion comparing their results with those obtained in other animal models of RP such as P23H mice and rats or Royal College of Surgeon (RCS) rats.
Author Response
Answers to Reviewer 1
We appreciate the comments made by the reviewers to our manuscript and we thank them for their time. These suggestions improve the quality and clarity of our manuscript. The manuscript has been thoroughly analyzed and comments for reviewers detailing the changes have been made. We have also highlighted the changes made to the manuscript in red. We hope that our answers to these comments match the level of the questions.
- Introduction: Considering that the microglia cells are an important part of the results, the authors should explain whit more detail the role of microglia in the early stage of retinitis pigmentosa (RP). Different authors have demonstrated that in RP animal model the microglial cell activation starts earlier that photoreceptor death ( doi: 10.3389/fnana.2017.00014; DOI: 10.1038/srep33356).
We agree with the reviewer and have written new information about the role of microglia in retinal degenerations. We have written in red rge new information. Now in the Introduction section it can be read as follows:
“In healthy retina, microglia can be found in the ganglion cells layer and in the inner and outer plexiform layers [12]. They have a branched shape with long, movable extensions that actively survey ocular environment [11]. In response to local injury, infection, or retinal degeneration, microglia morphology can change to an amoeboid shape and migrate to the outer retina [11, 12]. Glia also play crucial roles at various life stages. During development and maturation, microglia support neuronal survival by interacting with neurons, surveilling the retinal microenvironment with their extensions, regulating synaptic plasticity through phagocytosis, and maintaining synaptic structural integrity and function [13].
Microglia activation has been shown in numerous retinal diseases, including age-related macular degeneration (AMD), glaucoma, diabetic retinopathy (DR), uveitis, retinal detachment, and RP. [11]. In fact, infiltration of damaged retinal regions by microglia has been observed in mice RP models (rd1 and rd10 mice) and in post-mortem samples from RP patients [12].
During the progression of different retinal degenerations, and particularly those that are characterized by progressive photoreceptor death (such as RP)., microglia have a dual role, either causing distress or protecting photoreceptors and inner neurons by monitoring, secreting substances, and engulfing cellular debris [13]. Though the exact role of microglia remain uncertain under pathological conditions, it has been demonstrated an increase in microglial cell activation that coincided with the initiation of photoreceptor death in several retinal degeneration animal models [14]. Researchers have demonstrated proliferation of microglial cells in retinas from P23H-1 rats (carrying a rhodopsin mutation) and Royal College of Surgeon (RCS) rats (with pigment epithelium malfunction) while glial fibrillary acid protein (GFAP) over-expression was observed to begin later. [14].
Undoubtedly, microglia represent a promising therapeutic target for RD, understanding their precise functions in different pathological contexts is therefore needed [13].”
- Gao, H.; Huang, X.; He, J.; Zou, T.; Chen, X.; Xu, H. The roles of microglia in neural remodeling during retinal degeneration. Histology and histopathology, 2022, 37(1), 1-10.
- Di Pierdomenico, J; García-Ayuso, D.; Pinilla, I.; Cuenca, N.; Vidal-Sanz, M.; Agudo-Barriuso, M.; Villegas-Pérez, M. P. Early Events in Retinal Degeneration Caused by Rhodopsin Mutation or Pigment Epithelium Malfunction: Differences and Similarities. Frontiers in neuroanatomy, 2017, 11, 14.
- Material and methods: The authors should explain in more detail the control and experimental animal groups. To make it easier to understand the manuscripts for the readers the authors should indicate the n for each control and experimental group and explain better the control group.
We have written new information about the control mice and the number of animals used in each experiment. Now it can be read as follows:
“C3H or control mice belong to the strain C3Sn.BLiA-Pde6b+/DnJ, homozygous for Pde6b+. This C3H congenic strain lacks the retinal degeneration gene Pde6brd1 that is characteristic of C3H substrains and thus offers most of the strain characteristics of the C3H background but without the early onset retinal degeneration. The rd1 mice (C3H/HeJ) model of retinal degeneration were also used; these mice are homozygous for the retinal degeneration 1 mutation (Pde6brd1), causing blindness by weaning age. All mice were derived from the Jackson Laboratory colony (The Jackson Labs, Bar Harbor, ME, USA). Four mice per group were used for immunohistochemistry determinations and four mice per groups were used for western blot techniques.”
- Material and methods: The authors should describe in more detail how the microglial cells were quantified whether manually or automatically.? Specify in which areas of the retina the microglia cells have been quantified and whether they have always been the same in all the slices analyzed, because it has been shown that in animal models of RP microglia activation is not the same in the dorsal part of the retina as in the ventral part (DOI: 10.3389/fnana.2017.00014, doi: 10.4103/1673-5374.251204).
We agree and have described in detail quantification of microglial cells, that has been performed like the study suggested by the reviewer. Now, in the Material and methods section, it can be read as follows:
“Iba1 expression was assessed by quantifying the number of positive cells similarly to Di Pierdomenico et al [14]. Within each animal, three sagittal cross-sections containing the optic disk were chosen based on section quality. Three photographs (at 20x magnifica-tion) were captured for each section. The distance between the optic disc and the retinal periphery was measured and the three images were taken in each section at a distance equivalent to 50% of the length between the optic disk and the retinal periphery. The numbers of the different microglial (branched and ameboid) cells were subsequently counted in each photomicrograph according to their morphology. Image quantification, covering an approximate length of 500 micrometers, was conducted manually with the assistance of ImageJ software. These individual counts were pooled to calculate the average number of microglial cells per animal (four animals per age were analyzed, n=4).”
- Material and methods: Authors should specify the size/portion of the retinal that has been quantified (define arbitrary unit).
As suggested by the reviewer, we have now explained in detail how Iba1 positive cells have been quantified (see question number 3). We have also defined what is arbitrary unit in the legend of figure number 2.
- Material and methods: Authors should specify how the branching and morphology of microglia cells have been quantified.
The branching of microglia cells (i.e. number of branches or length of the branches) has not been quantified in this study. However, we have identified and quantified manually the number of ameboid or branched Iba1 cells manually according to their morphology. These aspects have been now explained in the Material and Methods section (see also answer to question 2).
- Results: Figure 1 and results of iba1+ cell quantifications: The iba-1 labelling shown in Figure 1 is very poor and does not allow a real quantification of the microglia cells present in the retina. Other authors, some also cited in this article (Zohu et al. 2017 DOI: 10.3389/fnana.2017.00077), show a significantly higher number of microglia cells in the retina of rd1 mice. For this reason, my suggestion is to double-check the quantifications of iba-1+ cells with images where the iba-1 labelling is cleaner, allowing the identification of all iba1 cells present in the retina.
We have tried to improve the quality of the image. In addition, the image is now in TIFF formar. We have also re-checqued the quantification of Iba1 cells and confirmed our results. However we have changed the discussion to try to clarify that our results are similar to the ones of Zhou et al. in 2017. In that study the researchers observed an increase in the number of microglial cells in degenerated retinas at post-natal day14. We have not observed any increase in this type of cells at post-natal day 11, but yes at post-natal day 17. This is perfectly compatible with the results observed by Zhou. Now in the Discussion section, it can be read as follows:
“We have also found an evident increase in marked microglia, both with an amoe-boid or branched shape, between PN11 and PN17. This increase, particularly in Iba1 cells of amoeboid morphology, was also shown by Zhou et al. in 2018, at PN14 and in the rd1 model. Furthermore, they demonstrated the protective effect of alpha-1-antitrypsin on neuroinflammation and degeneration at the retinal level in rd1 mice [32]. This same author published, a year earlier, a study on the evolution of microglia, in terms of number and morphology, in the rd1 retina at postnatal ages PN14, PN21, PN28 and PN180 [33]. These authors identified the peak density of microglia with amoeboid majority at PN14, decreasing at later ages. These results are similar to the ones obtained in this study, we have not studied PN14, but we have observed no changes at PN11 and an increase in total number of microglial cells, and particularly ameboid Iba1 positive cells, at PN17.”
- Discussion: It would be interesting if the authors included a paragraph in the discussion comparing their results with those obtained in other animal models of RP such as P23H mice and rats or Royal College of Surgeon (RCS) rats.
We have added information to the Discussion section regarding the results obtained in other studies in other animal models.
For example, now it can be read as follows:
“All studies related to microglia in rd1 retina, mentioned so far, establish in one way or another a close relationship between microglia and the degenerative process of photoreceptors, which takes place in ONL of the RP retina. This has also been demonstrated in other non-murine animal models that have described, in rat models, the activation and migration of microglia towards the ONL, from the beginning of the degeneration of photoreceptor cells [43]. Our results are also in agreement with those obtained by He et al. [44], who used the Royal College of Surgeon (RCS) rat model characterized by a recessively inherited mutation in the receptor tyrosine kinase gene Mertk, resulting in secondary degeneration of photoreceptors between postnatal days 20 and 60. Although in our rd1 mouse model, the retinal degeneration process is faster, the trend of increased Iba-1 cells in the retina is similar in comparison with their respective control group.”
“All studies related to microglia in rd1 retina, mentioned so far, establish in one way or another a close relationship between microglia and the degenerative process of photoreceptors, which takes place in ONL of the RP retina. This has also been demon-strated in other non-murine animal models that have described, in rat models, the ac-tivation and migration of microglia towards the ONL, from the beginning of the de-generation of photoreceptor cells [43]. Our results are also in agreement with those obtained by He et al. [44], who used the Royal College of Surgeon (RCS) rat model characterized by a recessively inherited mutation in the receptor tyrosine kinase gene Mertk, resulting in secondary degeneration of photoreceptors between postnatal days 20 and 60. Although in our rd1 mouse model, the retinal degeneration process is faster, the trend of increased Iba-1 cells in the retina is similar in comparison with their respective control group.”

Reviewer 2 Report
This paper explores an interesting area and it's written well. But, there's a big issue with your methodology - specifically about how many subjects are in each group. You've mentioned that there are at least three in each group. But, to run a proper statistical analysis, you need at least five independent observations per group.
With the number of subjects you have at the moment, it's impossible to work out the distribution of your data. This means you can't use traditional statistics. You have a few options to address this:
1) Increase the number of subjects to at least five per group. This will make your statistical analysis more reliable.
2) Don't do a statistical analysis. Instead, you can present your results as preliminary findings. Make sure you mention this limitation in your discussion.
3) Use more complex statistical methods such as bootstrapping or permutation tests. These aren't as widely used and aren't always reliable.
Please consider these points to improve your study.
Author Response
Answers to reviewer 2
We appreciate the comments made by the reviewers to our manuscript and we thank them for their time. These suggestions improve the quality and clarity of our manuscript. The manuscript has been thoroughly analyzed and comments for reviewers detailing the changes have been made. We have also highlighted the changes made to the manuscript in red. We hope that our answers to these comments match the level of the questions.
- With the number of subjects you have at the moment, it's impossible to work out the distribution of your data. This means you can't use traditional statistics. You have a few options to address this:
1) Increase the number of subjects to at least five per group. This will make your statistical analysis more reliable.
2) Don't do a statistical analysis. Instead, you can present your results as preliminary findings. Make sure you mention this limitation in your discussion.
3) Use more complex statistical methods such as bootstrapping or permutation tests. These aren't as widely used and aren't always reliable.
The number of animals used in our study was 4 in each group and each age. We have added this information in each of the figure legends.
We have also changed the Discussion trying to make it clear that our results are preliminary and that one of the limitations of the study is the number of mice in each group:
“In the rd1 mice, rod death occurs during the second and third postnatal week and the death of the cones takes place after the fifth week and continues for several months [27]. Thus, at approximately PN28, it can be said that the degeneration of the cones begins. At this postnatal age, our data showed that the macroautophagy marker Atg7 is significantly decreased in the rd1 group, compared to the control group (n=4 in each group) (Figure 6). These preliminary results may confirm others that affirm that macroautophagy process is detected only for 24 hours at the beginning of the degeneration of the cones in the rd1 mice, prior to chaperone-mediated autophagy (CMA) [44].
CMA is altered later in rd1 mice, at the ages at which cone degeneration occurs [44]. This agrees with the preliminary results of this study, at the two later ages studied, PN35 and PN42, the results reveal statistically significant alterations in LAMP2A, a marker of chaperone-mediated autophagy (CMA) in the retina of rd1 mice. However, we have observed that LAMP2A is decreased at PN35 and increased at PN42 (Figure 7). The increase in CMA observed in rd1 retina may be a compensatory process. CMA activation has been described in vitro in macroautophagy-deficient cell lines [45], as a compensatory process. Specifically in the retina, a decrease in the Atg5 marker has been classified as the cause of a compensatory mechanism in the form of CMA activation [46].
Therefore, we can preliminarly conclude that during the early phases of life and retinal ldegeneration (between PN11 and PN28), in the retina of the rd1 animal model, there is an alteration in microglia and the macroautophagy cycle. Subsequently, the CMA, as stated because LAMP2A expression, is decreased and later appears activated as a compensatory mechanism (Figure 8). The results obtained in this study are limited because the number of animals in each group was four and new studies should be performed to confirm them.”

Reviewer 3 Report
This manuscript by Martinez-Gonzalez et al investigated the sequential changes of inflammation and autophagy proteins in the rd1 mice, one of well-established animal model of RP. The content of this report is novel. I have some comments that need to be addressed first:
1. Antibodies are used in this study, the purchase of these antibodies from commercial source doesn’t guarantee the specificity of these antibodies, the authors need to at least cite some references to mention the specificity of antibodies employed, or the specificity has been validated in their previous studies including using KO mice or other methods..
2. The authors used the immunofluorescence (IF) labeling of Iba1 for determining the upregulation of Iba1 protein while using WB to monitor the autophagy proteins. From my understanding, the IF was widely used for determining the cellular localization of the proteins. Is there any specific reason to use IF but not WB to check the Iba1 expression?
3. Did the authors used both genders (male and female) or only male or female mice in this study, if so, please add it in the context.
4. For WB, please add which method was used to determine the extracted protein concentration?
5. I believe the authors used “anti-beta-Actin” not “anti-beta-Actine” antibody, please verify and change “Actine” to “Actin”.
6. Please add the predicted molecular weight of antibody targeted antigens in table 1.
7. Please use “Iba1” or “Iba-1” consistent.
8. If possible, for Iba1 IF labeling, please mention or show the absence of detection of Iba1 with primary antibody omission and only use secondary antibody, because using Alexa-488 secondary antibody at 1:200 is relatively high or may show some background labeling.
9. From line 123, please check “and cones are the cells that dye” or “and cones are the cells that die”.
10. From line 233, please change “ in the retina od the rd1 mice” to “ in the retina of the rd1 mice”.
Thank you!
The Quality of English language is OK to me.
Author Response
Answers to Reviewer 3
We appreciate the comments made by the reviewers to our manuscript and we thank them for their time. These suggestions improve cthe quality and clarity of our manuscript. The manuscript has been thoroughly analyzed and comments for reviewers detailing the changes have been made. We have also highlighted the changes made to the manuscript in red. We hope that our answers to these comments match the level of the questions.
- Antibodies are used in this study, the purchase of these antibodies from commercial source doesn’t guarantee the specificity of these antibodies, the authors need to at least cite some references to mention the specificity of antibodies employed, or the specificity has been validated in their previous studies including using KO mice or other methods.
We have cited some references in the Material and Methods section to confirm the specificity of the antibodies employed. Now it can be read as follows:
“The Wako anti-Iba-1 antibody was chosen for immunohistochemistry due to its proven specificity in mouse retina, as demonstrated in the study by Zhang et al. [32]. Similarly, the antibodies used in this study to determine the autophagy status in the retina via western blot were also confirmed in the literature to be specific for their use in mouse retina (Anti-Beclin-1 [33], Anti-Atg5 [34], Anti-Atg7 [35], Anti-LC3 [36] and Anti-LAMP-2A [37]).”
- Zhang, M.; Zhong, L.; Han, X.; Xiong, G.; Xu, D.; Zhang, S.; Cheng, H.; Chiu, K.; Xu, Y. Brain and Retinal Abnormalities in the 5xFAD Mouse Model of Alzheimer's Disease at Early Stages. Frontiers in neuroscience, 2021, 15, 681831.
- Lee, S. J.; Kim, H.P.; Jin, Y.; Choi, A.M.; Ryter, S.W. Beclin 1 deficiency is associated with increased hypoxia-induced angiogenesis. Autophagy, 2011, 7(8), 829–839.
- Ross, B. X.; Jia, L.; Kong, D.; Wang, T.; Hager, H.M.; Abcouwer, S.F., Zacks, D.N. Conditional Knock out of High-Mobility Group Box 1 (HMGB1) in Rods Reduces Autophagy Activation after Retinal Detachment. Cells, 2021, 10(8), 2010.
- Giansanti, V.; Rodriguez, G.E.; Savoldelli, M.; Gioia, R.; Forlino, A.; Mazzini, G.; Pennati, M.; Zaffaroni, N.; Scovassi, A.I.; Torriglia, A. Characterization of stress response in human retinal epithelial cells. Journal of cellular and molecular medicine, 2013, 17(1), 103–115.
- Gupta, U.; Ghosh, S.; Wallace, C.T.; Shang, P.; Xin, Y.; Nair, A.P.; Yazdankhah, M.; Strizhakova, A.; Ross, M.A.; Liu, H.; Hose, S.; Stepicheva, N.A.; Chowdhury, O.; Nemani, M.; Maddipatla, V.; Grebe, R.; Das, M.; Lathrop, K.L.; Sahel, J.A.; Zigler, J.S.; Sinha, D. Increased LCN2 (lipocalin 2) in the RPE decreases autophagy and activates inflammasome-ferroptosis processes in a mouse model of dry AMD. Autophagy, 2023, 19(1), 92–111.
- Trachsel-Moncho, L.; Benlloch-Navarro, S.; Fernández-Carbonell, Á.; Ramírez-Lamelas, D.T.; Olivar, T.; Silvestre, D.; Poch, E.; Miranda, M. Oxidative stress and autophagy-related changes during retinal degeneration and development. Cell death & disease, 2018, 9(8), 812.
- The authors used the immunofluorescence (IF) labeling of Iba1 for determining the upregulation of Iba1 protein while using WB to monitor the autophagy proteins. From my understanding, the IF was widely used for determining the cellular localization of the proteins. Is there any specific reason to use IF but not WB to check the Iba1 expression?
The immunohistochemistry technique was chosen to determine the expression of Iba-1, as it allows us to identify the location of the Iba-1 positive cell somas, their morphology, and their degree of activation. This information has been added to the Material and Methods section.
- Did the authors used both genders (male and female) or only male or female mice in this study, if so, please add it in the context.
We have used male and female mice in the study. We have added this information at the begining of the Material and Methods section.
- For WB, please add which method was used to determine the extracted protein concentration?
We have used the Bradford method to determine protein concentration. This information has been now written in the Material And Methods section.
- I believe the authors used “anti-beta-Actin” not “anti-beta-Actine” antibody, please verify and change “Actine” to “Actin”.
We apologize for the mistake. Now it can be read “Actin” instead of “Actine”.
- Please add the predicted molecular weight of antibody targeted antigens in table 1.
We have writen the predicted molecular weight of antibodies to the Table 1 of the manuscript as suggested by reviewer.
- Please use “Iba1” or “Iba-1” consistent.
We apologize for the mistake and changed “Iba-1” to “Iba1” throughout the manuscript.
- If possible, for Iba1 IF labeling, please mention or show the absence of detection of Iba1 with primary antibody omission and only use secondary antibody, because using Alexa-488 secondary antibody at 1:200 is relatively high or may show some background labeling.
We have added a suplementary figure which shows a retinal image only using the secondary antibody (negative control) and showing the absence of Iba1 detection.
- From line 123, please check “and cones are the cells that dye” or “and cones are the cells that die”.
We apologize for the mistake and have corrected the sentence as suggested by reviewer.
- From line 233, please change “ in the retina od the rd1 mice” to “ in the retina of the rd1 mice”.
We apologize for the mistake and have corrected the sentence as suggested by reviewer.

Round 2
Reviewer 1 Report
After the correction made by the authors, the manuscript has improved significantly and fulfills the jounal requirements for publication.
.